# 'We the People': Demarcating the Demos in Populist Mobilization—The Case of the Italian Lega

Oliver Schmidtke 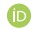

Centre for Global Studies, Department of Political Science, University of Victoria, Victoria, BC V8S 1L4, Canada; ofs@uvic.ca

**Abstract:** This article is a theoretically guided and empirically based analysis of how populist movements invoke the notion of the 'people' as a cornerstone of their political mobilization. While the confrontation between the virtuous 'people' and the unresponsive elites speaks to how populism challenges established political actors and institutions, the actual meaning of who the 'people' are and what they represent is shifting and often driven by strategic considerations. Analytically the article investigates the distinct ways in which nationalism and populism conceptualize and politically mobilize the notion of the 'people'. Empirically it focuses on the Italian League and engages in a discourse analysis of its political campaigns over the past 30 years. Based on this textual analysis of political campaigns, the article sheds light on how the reference to the 'people' has been employed as this political actor has transformed from a regionalist party advocating for autonomy in Northern Italy to one taking up the role of a populist-nationalist party at the national level. This case study allows the author to make a generalizable hypothesis about the nature of identity politics promoted by populist actors and the way in which the invocation of the 'people' and their alleged enemies is a pivotal political narrative that opens and restricts opportunities for political mobilization. This interpretative approach also allows for a more concise conceptual understanding of the affinity that right-wing populists demonstrate toward nativist ideologies.

**Keywords:** populism; nationalism; regionalism; italy; democracy; party politics; Lega

## 1. Introduction

'We are the people', a slogan originally used by the opposition against the German Democratic Republic's Communist regime in 1989, has been adopted by the right-wing *Alternative for Germany* (Volk 2020). Similarly, populist actors in Western democracies challenge established political elites by invoking the image of a people whose rights are violated and whose political voice is suppressed. They claim to speak on behalf of the 'people', and a cornerstone of the populist playbook of political mobilization is the way in which these populist groups claim to represent 'the people's' genuine interests which, they argue, have been neglected by the 'establishment' (Agnew and Shin 2019; De La Torre 2013; Espejo 2017). Much of the populist appeal rests on the claim of reinstating the proper place of the 'sovereign' people—depicting the 'sovereign' people as the site of ultimate democratically legitimated authority and, at the same time, as betrayed by the very institutions that are supposed to guard their sovereign rights (Spruyt et al. 2016). Populist actors mobilize the demand for 'popular sovereignty' with the explicit intent of challenging established decision-making procedures in liberal democracies (Urbinati 2019).

Claiming to be the authentic vox populi, the direct and unfettered voice of the people, gives populists' politics its sense of urgency and emotional vitality (Salmela and Scheve 2017). The regular public staging of the 'people' (both physically at rallies and demonstrations, as well as virtually across social media[1] platforms) provides the rhetorical and identity-driven fuel for the dramatized contrast to the political elite or establishment (Jagers and Walgrave 2007).[2] Mudde notes that the term 'populism' can be described as

"an ideology that considers society to be ultimately separated into two homogeneous and antagonistic groups—'the pure people' versus the 'corrupt elite', and which argues that politics should be an expression of the volonté générale (general will) of the people" (Mudde 2004, p. 543). Taggart (2000, p. 95) speaks of a 'heartland' that is central to the populist mode of mobilization as a place 'in which, in the populist imagination, a virtuous and unified population resides'. This form of contentious politics—its reliance on direct political action, a strong mobilizing collective identity, and charismatic leadership—is the constitutive mark of populism (De La Torre 2013; Moffitt and Tormey 2014; Mudde and Kaltwasser 2012).

Considering Mudde's and Stanley's claim that populism should be considered a 'thin-centered ideology' (Mudde 2007, p. 33; Stanley 2008), one could argue that the populist invocation of the virtuous and sovereign people can be seen as yet another facet of the rhetorical arsenal that such political actors use strategically and without substantial ideological principles. From this perspective, the populist assertion of representing the 'people' authentically could be interpreted as mere spectacle. In contrast, this article argues that the prominent reference to the 'people' is a fundamental feature of the populist political project that speaks to the nature of its mobilization and—at least in its right-wing version—its affinity toward nativist ideologies (Betz 2017). The key hypothesis of this article is that the way in which the foundational reference to the 'people' is articulated and mobilized for strategic purposes has itself a significant effect on the populist political agenda. In particular, the mode of framing the Us-versus-Them binary is a prime indicator of how populist actors situate themselves with regards to a nationalist political rationale and its exclusionary logic of bordering the 'people' in ethno-cultural terms.

The reference to the 'people' as a constitutive component of populism's political identity and the way in which it shapes the political options available to populist actors will be explored throughout this article using the example of the Italian League (henceforth simply: Lega). To focus on this political actor as a case study is illuminating, as the Lega has proven to be a political chameleon over the past three decades that has effectively combined elements of a far-right, anti-immigrant ideology and a populist anti-elitism. In the 1990s, the Lega Lombarda-Nord saw a meteoritic rise as a regionalist-populist opposition to the national political elite that was marred in corruption scandals across the political spectrum. In the wake of extraordinary electoral successes, the Lega Nord joined a series of national coalition governments (initially as a junior coalition partner under Silvio Berlusconi) advocating for greater regional autonomy, if not the independence of 'Padania' (encapsulating Northern and parts of Central Italy; see: Cachafeiro 2017). More recently, under the leadership of Matteo Salvini, the Lega has taken on the role of a nationalist, right-of-center opposition party seeking to expand its electoral support to the south of the country. Empirically, this article explores two main issues: First, it investigates the Lega's transformation from its early stages as a regionalist-populist party in the North to its current iteration as a national party on the right. Second, the article analyses the strategic reliance on populist and nationalist tropes of political mobilization reflected in how the Lega depicts 'people' in its political campaigns.

This text starts with a short theoretical reflection on the relationship between populism and nationalism with a view to how the reference of the 'people' is mobilized and politically used. I will then proceed to analyze the framing of the Lega over three different periods reflecting its transformation from a regionalist party to one that seeks to occupy the role of the primary nationalist party on the right. The investigation into this political actor allows me to draw some general insight into the way in which populist parties employ the mobilizing appeal to the 'people' and how this narrative opens and restricts opportunities for their political mobilization. The hypothesis that I seek to develop based on the Lega as a case study is that, compared to nationalist modes of political mobilization, populism can effectively redefine its notion of the 'people' in response to emerging political opportunities. Nevertheless, at the same time, strategically using a particular meaning and identity

attributed to the 'people', populist actors redefine the scope within which they can mobilize their anti-elitist political creed.

## 2. Defining the 'Voice of the People': Populism and Nationalism

Populism's ideological ambiguity and popular appeal make this political force an intellectually fascinating, albeit theoretically challenging, subject of study. The conceptual uncertainty is rooted in the versatility of the claim to represent the interest of ordinary people in a direct and authentic manner. Populism is a mode of engaging in politics that is not exclusive to a particular ideological position or type of political actor. The chameleon-like nature of populism has made it problematic to locate populist actors on the left–right axis and to define their core political aspirations (Mudde and Kaltwasser 2012).

If indeed populism is best conceptualized as a thin-centered ideology and mode of mobilization, it is critical to shift the analytical focus on the claims constituting its popular appeal in the current political climate. At the core of right-wing populist political strategy is the reference to the 'people' as a collective that they depict as deprived by the elite with a view to its collective identity and socioeconomic interests (Canovan 2002). The charismatic leader regularly claims to articulate the direct 'voice of the people' untamed by procedural rules associated with liberal democracy. Given the centrality of the 'people' as justifying the populist cause and the mode of conducting politics, populism needs a tangible and emotionally charged sense of the community on which it claims to rely on as its raison d'être. The rallies and manifestations of populist actors are no coincidental manifestation; they speak directly to the significance attributed to the dramatized depiction of the community of regular people (Schmidtke 2021). Populists draw on the sense of unity and cohesion staged at mass gatherings. It is here where the 'imagined community' gains a fleeting manifestation; the *demos* take on a theatrical existence sanctioning the 'people' and, by virtue of the latter, its populist leader.

It is in this respect that the affinity between right-wing populism and nationalism becomes apparent. The discourse of both revolves around the notion of the sovereignty of 'the people'. The mobilizing power of identity politics associated with populism and nationalism relies centrally on the valorization of the 'people' as a relatively homogenous political community that needs to be defended against its internal and external enemies. Given the inherently binary relation articulated by forms of collective identity, this reliance on a positive depiction of the own group is normally combined with the degeneration of the 'other'.

Canovan (2002, p. 34) points out that populist politics conceives of the 'people' in terms of a homogenous group with a clearly distinguishable sense of identity and shared interests articulated in a common will—a 'volonté générale'. In the same vein, Taggart underlines that "populist rhetoric uses the language of the people not because this expresses deeply rooted democratic convictions about the sovereignty of the masses, but because 'the people' are the occupants of the heartland and this is what, in essence, populists are trying to evoke" (Taggart 2000, p. 95). For Mudde and Kaltwasser (2012), the notion of 'the people' could imply the people 'as sovereign', 'as nation', and 'as the common people'. As will become apparent in the subsequent case study of the Italian Lega, populist actors from the right tend to combine these three reference points in depicting the 'people' in a politically dynamic and productive, albeit not necessarily consistent, fashion.

With focus on the 'people' being a key feature in populist parties' political mobilization, this article builds on the prominent role of identity politics that recent scholarship on populism has identified as a key component of this actor's political success (Hawkins et al. 2017; Spruyt et al. 2016). In this respect, the sense of a virtuous and homogenous community is not only the emotional and mobilizing way of defining a collective agent for populist actors (Salmela and Scheve 2017). Rather, the plea to defend the genuine ideas of this community is at the core of the populist anti-elitist challenge to the political status quo. This form of identity politics has been instrumental in transcending the traditional

left–right cleavage and in providing an effective political voice to prevalent socioeconomic grievances (Marchlewska et al. 2018; Noury and Roland 2020; Norris and Inglehart 2019).

However, it is worth noting that the populist and nationalist traditions of referring to the 'people' show distinct features. Both forces engage in exploiting this binary, but they tend to define the 'enemies of the people' differently. In the scholarly discussion on comparing the discourses of both political forces, populists are depicted as operating based on a vertical axis pitching ordinary citizens against unresponsive elites, while nationalists are portrayed as promoting a horizontal sense of the people as a politically or culturally bounded community.[3] The emphasis of populists is primarily on the deprivation that a particular political community claims to suffer from (predominantly framed in terms of a violation of 'sovereign' rights). In contrast, nationalists put the shared ethno-cultural identity at the center of their political mobilization and demand its protection from 'corrupting' outside forces (Calhoun 1993). Gellner spoke of the innate tendency of nationalism to promote 'mono-culturalism' (Gellner 1997), the commitment to nourishing and protecting the ethno-cultural identity of the dominant group. The foundational idea of national homogeneity can also rely on nativism, an ideology that is aggressively directed toward those who are seen not to be legitimate members of the native (national) group (Hervik 2015; Mudde 2007; Wodak 2013). In this case, the borders between the two groups are unambiguous, or as Bauman (2001, p. 12) observes, "it is crystal-clear who is 'one of us' and who is not, there is no muddle and no cause for confusion".

As Brubaker has argued convincingly, these dimensions of invoking the 'people' normally intersect in the practice of both political movements (Brubaker 2020). In populist political narratives, the politically potent reference to the 'people' points to both, people as those who have been deprived of their legitimate rights and people as a bounded community whose identity and interests need to be protected and nurtured.[4] For instance, the strong anti-immigrant rhetoric and insistence on—national—borders as the ultimate defense of the sovereign rights of the 'people' regularly shapes the political discourse of nationalists and populists alike. In this regard, I consider Brubaker's claim persuasive that "this strict conceptual separation cannot capture the productive ambiguity of populist appeals to 'the people', evoking at once plebs, sovereign demos, and bounded community." (Brubaker 2020, p. 44). Populists employ the nationalist allure of portraying people united as equals by cultural traits and a shared collective decision-making process. Nevertheless, in the discourse of right-wing populism, the issues of inequality and deprivation are regularly fused with an often belligerent notion of the community's identity and borders (Lamour and Varga 2020).

This collective identity is instrumental in turning the perceived social and cultural marginalization into a vehicle of political protest. Borrowing from nationalist ideologies, but being far more versatile in staging the defining characteristics of the 'people', populists articulate a yearning for belonging and a romanticized past when this identity was supposed to have been pure and untainted (Bell 2008). In populist rhetoric, the invoked notion of the 'people' as community is—far from being a territorially, linguistically, or ethnically defined nation—a *chiffre* to direct political anger and frustration. The 'Make America great again' slogan leaves deliberate ambiguity in defining a nation's interests and identity.[5] Its primary purpose is to fuel a form of agonistic politics whose driving force is the contestation of the status quo (see Kapoor 2002).

In sum, while traditional regionalist movements operate based on historical nationalism and are committed to preserving and revitalizing their own culture, populist actors are inclined to use a rhetorical discourse about them politically. Populists tend to employ the legitimizing symbols from the classical repertoire of nationalist ideas as a mobilizing political narrative rather than being committed to the central political goals of the traditional nationalist agenda.

### 3. The Italian Lega: From Regionalist to Nationalist Populism

The Italian League is an intriguing example of a political movement and later party that used elements of populist mobilization and nationalist identity building in a highly versatile and productive manner. In many respects, the Lega can be considered an early political manifestation of the recent wave of populist politics that would become a momentous disruptive force in party politics across Europe. In the 1990s, the political landscape in Italy, driven by a series of corruption scandals known as Tangentopoli and the collapse of Italy's Communist Party (the Partito Comunista Italiano), provided a fertile ground for a political force such as the Lega to rise and challenge the established parties and institutions (Tarchi 2015). The Lega began as a Northern-regionalist force in old Christian–Democratic strongholds (such as Lombardy and Veneto).

It is worth considering how the Lega transformed in pursuit of expanding its political reach and power from the regionalist, if not separatist Lega Nord/Lombarda in the 1990s, first to the Lega Nord, and finally to its current manifestation as the national Lega under the leadership of Matteo Salvini. In its first iteration under the leadership of Umberto Bossi (during the formative 1990s and the early 2000s), the Lega combined two elements that would make it distinct from traditional forms of regionalist politics (Schmidtke 1996). Newth speaks of the Lega's 'populist regionalism' (Newth 2019, p. 384). With one dimension of its political identity, the Lega borrowed from the playbook of traditional territorial politics claiming to guard the interest of Lombardy and, when it became apparent that their political message resonated more broadly, Northern Italy (Giordano 1999). The Lega advocated for deepening Italy's federalism, if not demanding independence for what was constructed as the ethno-cultural 'heartland' of the movement: Padania (covering approximately the whole of Northern Italy and parts of Central Italy).

Very much in line with traditional nationalist arguments, the Lega sought to base its political legitimacy and project in the imagined community of a homogenous Northern Italian people (Ruzza and Schmidtke 1993). Using the playbook of nationalist politics, the Lega employed an aggressive anti-immigrant political strategy to give popular meaning to the historically insubstantial notion of what a Northern ethno-cultural identity would be. The Lega forcefully nurtured a notion of the Northern people of Padania by demarcating it largely from immigrants portrayed as threatening the well-being and identity of the own community. In this respect, the anti-immigrant discourse and its xenophobic underpinnings have been a continuous and important component of the party's political discourse and popular appeal (see Abbondanza and Bailo 2018; Colombo 2013; Ignazi 2005; Richardson and Colombo 2013).

In its ambitions to become the dominant force in Italy's north, the Lega gradually abandoned the reference to ethnic identity as a politically essential orientation and replaced it with a binary coding of the political discourse. The new strategy polemically confronted the failure of the nation state and contrasted the parties that represented the state with—as did the Lega in one of their campaigns—the 'sincere, hard-working, and people-oriented' workers and producers from Padania. In the Lega's populist version of regionalism, forms of collective identity based on cultural attitudes replaced those promoting an ethnic or even primordial narrative of belonging (Schmidtke 1996). During this period of its political mobilization, the territorial reference point in the North was decisively shaped by cultural elements of community-building, focusing on an individualistic work ethic that set itself apart from the economically underdeveloped South. From a political point of view the reference to a culturally defined work ethic, which has a long tradition in northern Italy, made more strategic sense, as it allowed additional political goals to be legitimized than would be the case with an ethno-cultural identity. At the same time, it is worth noting that during this early phase the Lega also targeted Central Italy with its political project (invoking the sense of 'Centronord'). The attempt to define its—expanding—Northern identity was based on an aggressive demarcation only from the South painted as inferior in socioeconomic and cultural terms.

However, the way in which a territorially framed collective identity was popularized—a collective identity that could not build on any historical, cultural, or political administration foundation—points to the second basis of the Lega's political identity. Unlike traditional nationalist actors and their attempts to rely on a narrative of primordial cultural and ethnic traits, the Lega created the image of an anti-elitist force fighting Italy's capital and the political establishment it claimed to represent. The (in-)famous slogan 'Roma ladrona' (Rome as the 'thief') summarized the primary political messaging: Northern Italians were depicted to be at the mercy of a wasteful and incompetent national government whose resources were primarily generated by the hard-working people of the North. During this period, Bossi presented the Lega as the political spokesperson for the industrious and honest Northerner suffering from the incompetence or dishonesty of the Roman government.

The Lega's brand of populist regionalism made skillful use of the polemical juxtaposition of the authentic and uncorrupted regional community on the one hand and the corrupt professional politicians and selfish interests of the nation's center on the other. Here, the highly democratic, 'non-alienated' alternative, for which the regional political community claims to stand, comes into direct confrontation with the assumed democratic inaccessibility of national institutions. In this respect, the Lega presented its populist version of regionalism as a protest directed at the national center and its representatives; it fed off the widespread alienation and frustration with established politics in the country.

One recurrent and constantly central political claim of the Lega throughout this transformation has been the populist appeal to an emotionally charged notion of the 'people', defined less by their territorial boundedness than their collective sense of alienation from and not having a voice in the current political system. In this respect, a key element of the Lega's current political mobilization is its social media campaigns (Mazzoleni 2014 calls this phenomenon 'mediatized populism'). As Bobba (2019) demonstrates the emotional style of its messaging caters to the central components of the populist agenda, most notably the sense of the 'people' allegedly betrayed or neglected by unresponsive elites.

Reinterpreting what the territorial reference meant for the movement, the Lega was able to change the geographical boundedness of the 'people' (Giordano 2001). As new political opportunities presented themselves to expand the reach and electoral appeal of the Lega, the sense of the 'heartland' defining the collective identity and territorial reference point of the 'people' that this movement claimed to represent altered. Territorially, this political community grew and now signified a larger Northern Italian, Padanian 'We'. Still, throughout the first decade of the new millennium, the Lega sought to keep the fight against Roman centralism and the plea for more regional autonomy as a keystone of its political messaging. This regionalist agenda was an important component of justifying the collaboration in national coalition governments (primarily under Silvio Berlusconi).

An even more radical shift in defining the boundaries of the 'people' in whose name the Lega claimed to speak happened with a decisive shift in leadership. In 2013, Matteo Salvini took over as the leader of the Lega Nord, quickly steering the party toward what he framed as a 'reinvention'. This re-imagining of the Lega saw the emergence of a populist, right-wing political formation with appeal across the country. In pursuit of new opportunities to expand the influence and electoral base of the Lega even further, this party dedicated more attention of its campaigns on the Southern parts of the country. Programmatically, the Lega decided to drop the 'Nord' from its name and move away from deepening regionalism or federalism in Italy as its key public policy demand. The focus of its campaigns has decidedly shifted toward a nationalist agenda with a strong emphasis on border protection and an opposition to Europeanization or globalization. Over the past three to four years, this redefinition of the Lega's political identity has paid off electorally. In 2019, Matteo Salvini's Lega exceeded 30% of the vote in the election to the European Parliament and became the largest party in Italy. Similarly, the Lega has had some notable success in expanding its electoral support to the Center and South of Italy in recent regional elections (Vampa 2021). According to current polls, the Lega is in a three-way race for

the biggest party in Italian politics, competing with the post-fascist *Brothers of Italy* (more details below) and the center-left *Democratic Party*[6].

In the subsequent section, I will empirically examine how the political transformation of the Lega—its development from a regionalist party firmly entrenched in a Northern Italian heartland and culture to a national(ist) populist party—is rooted in a shifting rhetoric of who the 'people' actually are. The analytical focus on the way the 'people' are defined and bordered can help to elucidate what kind of transformation we have witnessed to the political identity of the Lega and what the structural effects of this transformation are.

## 4. Defining the 'Other'—The Changing Political Discourse of the Lega

If indeed populist movements are not bound by a classical nationalist sense of the people's collective identity and tend to operate—at least rhetorically—on a rather inclusive notion of the sovereign people (Espejo 2017), it worth investigating how populists claim to represent the genuine 'sovereign people' in their actual political campaigns. Theoretically, this empirical analysis is based on two considerations: First, collective identities are essentially defined by the binary between 'Us' and 'Them'; the identity of the own community becomes specific only by what it is not. Second, the political mobilization of populist actors depends on projecting a plausible notion of who does not belong to or threaten the integrity of the 'people'. Thus, it is most promising for understanding the way populists provide meaning to the notion of the 'people' to focus on those who are deemed the 'Non-We', the threatening Other, or the enemy of the 'people'.

Applying this conceptual understanding, I conducted a qualitative study of party programs/manifestos, parliamentary responses, speeches from key party politicians, debates at party congresses, and social media campaigns (where applicable) during three distinct phases of the Lega's history (three-year periods each): 1995–1998, 2008–2011, and 2017–2020[7] I then coded these documents according to the following ways of depicting the 'people' through the lens of the 'Other'[8]. As outlined in Table 1, the coding system focused on distinguishing whom the documents in question portrayed as opposed, threatening, or hazardous to the 'people'.

**Table 1.** Modes of defining the 'Other' in Lega's discourse on the 'people'.

| Type of Coding the 'Other' | Defining Features |
| --- | --- |
| Immigrants/refugees | Reference to an ethnically or culturally defined immigrant or refugee depicted as the non-national other; modes of inclusion and exclusion demarcated by identity markers relating to cultural traditions/values, forms of ethnic belonging, or ancestry. |
| Southern Italians | Reference to those coming from or residing in the South of Italy[9] |
| State bureaucracy/political elites | Reference to political elites (primarily associated with the traditional party system of state bureaucracy). |
| Economic elites/big corporations | Reference to economic elites (regularly framed in terms of social inequality) and big corporations exerting power over the economy or society at large. |
| European Union | Reference to the European Union as an institution, set of policies, or the project of European integration as a whole. |

Adopting an interpretative research approach, the qualitative textual analysis identified one of the five codes providing the dominant narrative for how the people and their adversaries are depicted in each of the speeches or statements. Documents were coded for one dominant code, depending on the relative importance of these frames in organizing the narrative meaning of the text. To assure intercoder reliability, the first set of textual units were coded several times by different researchers. As a result of this experience, the categories described in Table 1 needed to be adjusted slightly for analytical clarity. In the subsequent analysis of the textual material, it proved to be relatively unproblematic to identify a dominant mode of stipulating the 'Other' that, in turn, was framed as opposed to the genuine 'people' in whose name the Lega claimed to speak.

If invoking the 'people' and claiming to be their voice, then the forces that threaten to deprive said 'people' constitute a pivotal role in the communication campaigns of the Lega. As Figure 1 shows, during its initial phase in the 1990s, the Lega put the emphasis squarely (over 40%) on the political elite as those representing the menacing counterpart to the virtuous 'people'. As briefly discussed above, the emphasis of this depiction of the 'enemy of the people' was directed at those representing the national parties, political institutions, and the Roman bureaucracy. The Lega's campaigns rested on the mobilizing antagonism between the virtuous, hard-working citizen from the North and the 'corrupt' Italian capital. Even when Umberto Bossi was elected to Parliament, he would publicly stage his disgust and promise his followers that he would return to the North/Padania on every occasion to avoid 'contamination' by the Roman elites and culture.

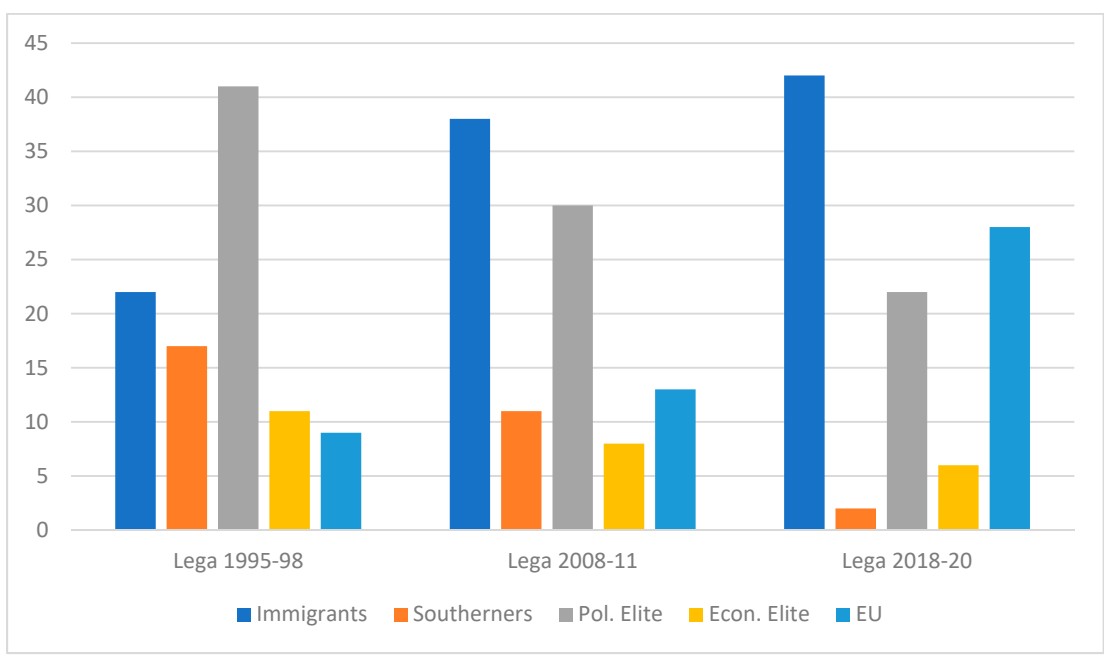

**Figure 1.** Defining the 'Other'—the Lega's framing of the threats to the 'people' (in %).

The other two components of the Lega's strategy of demarcating the 'people' were a persistent anti-immigrant rhetoric and an opposition directed at the 'Southerners' in Italy. Initially, the Lega popularized the nascent notion of a Northern Padania with reference to a distinct (and threatening) cultural identity of Southern Italians. From the beginning, the Lega combined a welfare chauvinism (pleading to keep resources and taxes in the region) with a culturally coded sense of those posing a threat to or being the antithesis of the 'people' in ethnically or culturally framed identity terms. Immigrants have always been a target of negative campaigning with a view to the alleged drain of resources and threat to a homogenous sense of who the 'people' of the North or Padania were supposed to be. Similarly, the focus on the Southerners underlined the notion of a Northern distinctiveness and lingering menace of losing this foundational collective identity.

At the same time, the belligerent attitude toward the Italians in the South was instrumental in socioeconomic conflicts, which according to the Lega, could only be addressed by a deepened federalism or regional independence. The Lega's initial 'regionalism' was essentially a protest against the 'meridionalization' (creeping takeover by the South) of the Roman government, which would be characterized by a clientelist, mafia-like practice that supposedly shaped the south. The Lega skillfully used the strong resentment against the South in Northern Italian society for populist mobilization. In contrast, economic elites and the European Union received relatively little attention in framing who the 'people' were pitched against (similar findings for the German context can be found in: Schmidtke 2020). This finding is all the more informative given that the Great Recession of 2008, and the experience of growing social inequality in Italian society shaped this period.

It is worth noting that, throughout its transformation from a regionalist actor to a right-wing nationalist opposition party, the Lega kept the anti-immigrant rhetoric as a constant feature in its political mobilization campaigns (Abbondanza and Bailo 2018). From the beginning, the immigrant has been the politically useful depiction of the 'Other' as an interpretative lens to portray the ills of society in terms of a simplistic result of outsiders claiming resources and privileges. In this regard, the Lega employed the political underpinning of exclusionary nationalism in a regionalist context (Mazzoleni and Ruzza 2018).

Regardless, while traditional regionalist movements operate based on historical nationalism and are committed to preserving and revitalizing their own culture, populist actors tend to use a rhetorical discourse about them politically. Populist actors politically used the legitimizing symbols from the classical repertoire of nationalist ideas to mobilize the confrontation of the nation state without describing its central political goals. Very much in line with this narrative of the 'Italian people' and its alleged adversaries is the relative absence of the focus on the Southerners or the economic elites in the campaigns of the Lega. The former would have—from a strategic stance—undermined the party's ambitions to expand into the center and in particular the South of the country (tapping into the historically engrained sense of being forgotten by the political elite in Rome). In addition, downplaying the economic elites (big companies, multi-nationalists) as the group that threatens the well-being and integrity of the 'people' reflects a long-term trend of the Lega slowly abandoning their anti-elitist impetus (at least with respect to the powerful economic elites[10]). In the most recent iteration of the Lega's populist mobilization, the anti-elitist impetus in delineating the 'people' is significantly downplayed (in particular with a view to socioeconomic elites) and replaced by a more direct affirmation of the 'people' identity in cultural terms.

In this regard, it is worth underlining how prominent the issue of immigration has become in the Lega's 'defense of the people' (Galbo 2020). During the last period analyzed here, the Lega put the emphasis on immigrants and the European Union as they threaten to compromise the well-being of the 'people's' community. The negative demarcation from immigrants took a particularly pronounced role in the political campaigns of the Lega. While the height of the so-called 'refugee crisis' was some years before the period under investigation, the issues of border controls, unregulated migration, and the impact of newcomers on Italian society were still a major focal point of public debates. In this respect, the Lega took advantage of a political opportunity that shapes Italian society and politics in important ways: The massive inflow of irregular migrants from Northern Africa has created a deep sense of vulnerability and frustration over the perceived lack of proper support and burden sharing from other EU member states. The Lega reacted to this issue and the ensuing political controversy by emphasizing its anti-immigrant rhetoric and the need to protect the 'people' from unwanted newcomers. The 'people' are now forcefully constructed as a national community whose borders are compromised and that is overwhelmed by the influx of foreigners (issues of crime, declining living standards, etc.).

The Lega employs a similar narrative of protecting the 'people' from outside threats and interventions with respect to the European Union and the project of European inte-

gration. While the role of the EU in defining the 'people' and articulating their interests used to be minimal in the late 1990s, the binary national versus European community has taken on a more pronounced role in the Lega's communication strategies. On many occasions, these images of the threatening 'Other' merged. The EU was regularly portrayed as the agency facilitating irregular immigration and imposing an undue burden on Italy. The emphasis on the political elite remained pronounced during the period under the leadership of Salvini, while Southerners and the economic elite lost even more of their influence on the narratives constructing the 'people' and highlighting the hazards they are facing (Richardson and Colombo 2013).

Considering the evolution of the Lega's political identity, it becomes clear how the departure from a regionalist party, invested in promoting the political capacity and collective identity of Northern Italy, has been legitimized and popularized by a substantially transformed notion of the 'people'. This shift in the Us-versus-Them binary allowed the Lega to re-define its political ambition as it was determined to become the dominant national party on the right. The 'people' are now primarily defined within the rationale of a classical national(ist) framework built on an aggressive demarcation toward immigrants or refugees and the European Union (Brunazzo and Gilbert 2017; Castelli Gattinara 2017). According to the current political formation of the Lega, the threats to the 'people' come primarily from outside of the country, implying a strong need for protecting borders and pushing for national sovereignty (see for a similar political agenda in the United Kingdom: Dennison and Geddes 2021; Schmidtke 2021). In a way, throughout its 30-year history, the Lega has come full circle: from a political actor claiming to protect the interests of the peoples of the North suppressed by the Italian nation-state to one promising to be the protector of this very nation from outside intervention.[11]

In its latest iteration under the leadership of Matteo Salvini, the Lega has largely adopted a logic of demarcating the people that is a defining feature of an exclusionary form of nationalism and ethno-cultural markers for defining identity and community. According to a detailed analysis of social media campaigns conducted by Albertazzi et al. (2018, p. 645), with Salvini taking over at the helm of the organization, the Lega embarked on reinventing itself as an 'empty form of nativist nationalism'.

Driven by the desire for expanding the base of loyalist and enhancing electoral base, the Lega has largely abandoned its regionalist agenda. Politically, this strategy is risky for two reasons. First, such a move is likely to alienate those supporters of Northern autonomy or independence who had found their political voice in the Lega. Second, the shift in the Lega's rhetoric concerning the 'people' they claim to represent and defend runs the risk of pitching the 'nationalized' Lega against other parties seeking to occupy the political space of the nationalist-populist right. In the Italian context, it is worth noting that, since its establishment as a nation-wide party in 2014, the Brothers of Italy (*Fratelli d'Italia*) have successfully expanded from its historic stronghold in Lazio in particular into the poorer Southern regions (De Giorgi and Tronconi 2018). Essentially, the right-wing populist party has replaced Berlusconi's *Forza Italia* in the South and, with its rising fortunes in the polls, has become a veritable competitor for the Lega (De Giorgi and Dias 2020). At the same time, the Lega is being challenged by what Hamdaoui (2021) calls a 'stylistic anti-populism' from the leftist Sardine movement (see also Newell 2020) that has staged a campaign of who the 'people' are based on local public manifestations in city piazzas.

## 5. Conclusions

For populist movements, the reference to the 'people' is a critical building block of providing their political project with narrative credibility and mobilizing vigor. To use the category of Mouffe (2018), the image of the virtuous 'people' that is deprived by irresponsive elites is a form of contentious politics that resonates in particular in times in which trust in established political institutions has eroded and a sense of insecurity is widespread (Mair 2013; Stavrakakis 2014). In times of highly complex relationships in politics and economy, the territorial community forms a politically effective frame of

reference in order to establish common political ground and a shared worldview. The notion of identity politics that the evocation of the 'people' allows to stage is one with a clearly defined enemy, an energizing grievance, and a sense of moral superiority of the own group. The Us-versus-Them binary is the central narrative on which populist political mobilization relies.

The case of the Italian Lega demonstrates how this reliance on the 'will of the people' is framed and who the 'Other' is, against whom the political aspirations of the claimed ordinary citizens are directed. In this respect, I interpret the 'people' in populist campaigns not only as an empty signifier that invites multiple or even contradictory interpretations of what constitutes the *demos* and what defines the boundaries of the political community. Rather, this practice in itself has a determining impact on the political project that populist forces advance. From a conceptual perspective, the reliance on a mobilizing notion of the 'people' can be interpreted as an essential component of the populist political identity that is employed strategically to exploit political opportunities and, at the same time, as a key feature of the populist political narrative that shapes its political preferences. In comparison to nationalist actors, populists can rely on a broad repertoire of mobilizing the notion of the deprived 'people' and its enemies. However, once a populist actor has embarked on a particular story of the 'people's' interests and identity, this narrative has its own structuring effects on the scope and mode of the populist political mobilization.

The transformation of the Lega's depiction of the 'people' underlines how the foundational populist narrative shapes political opportunities and ideological commitments of this type of political actor. By embarking on a depiction of the 'people' in line with a nationalist narrative, the original emphasis on an anti-elitist empowerment of ordinary citizens vis-à-vis the establishment has been relegated to a less important pillar of the Lega's contemporary political identity. The focus on the outside enemies and a narrative relying on the notion of a culturally homogenous 'people' have significantly replaced pleas for democratic and socioeconomic empowerment associated with the reliance on the 'volonté générale'. In this respect, the increasing prominence of nativist elements in the Lega's rhetoric is not accidental. This development reflects a new way of invoking the 'people' in the tradition of exclusionary nationalism to which right-wing populism has a structural affinity.

**Institutional Review Board Statement:** The study was conducted in accordance with the *Human Ethics Standard* of the University of Victoria (protocol code *19-0232*, approved 6 April 2020).

**Informed Consent Statement:** Informed consent was obtained from all subjects involved in the study.

**Data Availability Statement:** The data presented in this study are available on request from the corresponding author.

**Acknowledgments:** I would like to acknowledge that this article draws on research supported by the Social Sciences and Humanities Research Council of Canada.

**Conflicts of Interest:** The authors declare no conflict of interest.

## Notes

[1]　Similar to other populist actors, the Lega has been highly effective using social media and in particular Facebook for its political campaigns (see Engesser et al. 2017; Moffitt and Tormey 2014).

[2]　The vital claim of populist actors to represent the people depicted as virtuous and homogeneous is a common reference point in scholarly attempts to explain their ability to challenge the status quo (Albertazzi and McDonnell 2008; Barr 2009; De La Torre 2013, p. 202; Müller 2011). However, this reference is widely interpreted as a political rhetoric that is open to any ideological messaging.

[3]　See for example: Bonikowski et al. (2019); De Cleen (2017); De Cleen and Stavrakakis (2017).

[4]　See Eatwell and Goodwin (2018).

[5]　The Italian Lega provides a similar illustration for this argument. For further discussion see Albertazzi et al. (2018).

[6]　See https://www.politico.eu/europe-poll-of-polls/italy/ (accessed 18 April 2021).

[7]　　A total of 452 documents were analyzed and coded according to the competing binaries and modes of depicting the 'people' described in Table 1 (1995–98: p.138; 2008–11: 150; 2018–2020: 164). Documents were coded for the dominant narrative of how the 'people' were depicted through the binary Us-versus-Them imagery. I then calculated the occurrences for the three investigated periods as a percentage of the total number of coded textual units.

[8]　　Methodologically the analysis draws on discourse analysis in the tradition developed in social movement research: Gamson and Wolfsfeld (1993); Koopmans and Statham (1999).

[9]　　It is important to note here that, in particular, in the Lega's early political discourse, Southern Italians were regularly framed as culturally distinct from Northern and Central Italians moblizing a long-term prejudice against the South in Italian society (see Schneider 1998).

[10]　　In its initial phase, the Lega relatively strongly endorsed the frustration of smaller shopkeepers and business toward big companies and globalization in the North (see Zaslove 2011).

[11]　　The need for bordering that is regularly done with reference to the nation states also makes it difficult for populists in Europe to construct a transnational dimension of their movement (Moffitt 2017).

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
