# Peer review of "‘We the People’: Demarcating the Demos in Populist Mobilization—The Case of the Italian Lega"

_socsci, doi:10.3390/socsci10100351_

Round 1

Reviewer 1 Report

This is an interesting and well-written research. Before recommending its publications, however, I believe it can be improved in a number of ways. I hope that this list will be useful for both the author and the editors:

  • line 4: update the abstract after the revisions
  • line 23: reference for the first sentence
  • note no. 1: typo ("campaigns")
  • line 40: remove the first name
  • lines 55-58: if possible, add a reference
  • line 67: a bit more on the categorisation of the Lega: a far-right party as well as a populist one
  • lines 70-78: a bit more focus on how the Lega started as a regionalistic populist party, and how only recently it repackaged itself as a national party, thus finding the electoral success the author writes about
  • lines 71-72: as a junior coalition partner
  • line 86: a bit more on the structure of the paper, as well as its findings
  • lines 129-146: the author hints at some theoretical explanation for the electoral success of far-right populist parties, but the article needs a sub-section or at least a paragraph on such theories. Some of the sources the author cites provide useful insights to that end
  • line 172: better expression ("In a nutshell")?
  • line 185: space between "Communist Party" and the following bracket
  • lines 200-201: this reads a bit awkwardly, perhaps this is better "covering approximately the whole of Northern Italy and parts of Central Italy"
  • lines 212-215: I would add that the Lega was very attentive in its political suasion towards Central Italy (the oft-used "Centronord" idea), and, for a number of years, this has meant that only Southern Italy was targeted by the Lega's socio-ethnic insults
  • lines 270-272: update these data, if possible
  • section no. 3: this is an interesting and well-written section. However, I was surprised to see the migration issue mentioned only in passing, since it's one of the focal points of the Lega's communication strategy. Moreover, there is an apparent paradox when the Lega and migrants are concerned, as the Lega often "barked but didn't bite" in policy terms. The author deals with anti-immigration in the following section, but it is not recommendable to leave such a prominent issue dealt with only in the second half of this article. To that end, I recommend some important publications that address this (though there are many more): A) Bull, Anna Cento. "Addressing contradictory needs: the Lega Nord and Italian immigration policy." Patterns of Prejudice 44, no. 5 (2010): 411-431; B) Abbondanza, Gabriele. "Italy’s migration policies combating irregular immigration: From the early days to the present times." The International Spectator 52, no. 4 (2017): 76-92; C) Richardson, John E., and Monica Colombo. "Continuity and change in anti-immigrant discourse in Italy: An analysis of the visual propaganda of the Lega Nord." Journal of Language and Politics 12, no. 2 (2013): 180-202
  • section no. 4: there needs to be a sub-section or at least a paragraph on the methodology employed by the author, with relevant references and a justification for its use in this article (a footnote cannot suffice)
  • line 289: good, but please explain (footnote?) that, in the Lega's communication strategy, Southern Italians are distinct from Northern and Central Italians
  • line 305: as mentioned in the previous section... (I'm referring to the discussion I invite the author to make in the previous section)
  • line 322: appropriate graph with interesting content
  • lines 323-329: good, this will link well with the previous discussion on migration
  • lines 350-358: although it's a delicate topic to approach, it would be good to mention that Italy was facing the largest seaborne migration route of the world, and that this has had political, security, economic (3 billion euros a year, see Villa et. al.) and psychological implications that other parties (and indeed the EU as a whole) did not choose to (or were not able to) effectively address. This resulted into a huge political advantage for the Lega
  • lines 381-384: good point
  • line 385: make this a proper heading
  • line 388: remove the first name (Chantal)
  • lines 441-446: why are these lines all in italics?
  • section no. 5: An interesting conclusion, though I would invite the author to shorten it, and to resort to the stylistic conventions of social sciences (summarising premises, goals, means, findings, and implications of the article)

With the above addressed, this article will be ready for publication, and should provide interesting food for thought for scholars interested in populism, far-right parties, and the Lega.

Author Response

Thank you very much for your most helpful suggestions. I hope that I have addressed them all in a satisfactory manner. Most importantly, I added a section on the theoretical debate on the driving forces behind populism, discussed more fully the issue of migration in the Lega’s political mobilization, added a paragraph on the methodological approach of my empirical research and shortened/ tightened the conclusions. Responding to one other reviewer I also tried to elaborate more fully the generalizable finding of my article and what it contributes to the scholarly debate on populism.

Reviewer 2 Report

This is an excellent, well-written and clearly organized and argued paper.  I learnt quite a bit from the discussion as well as from the bibliography.  I look forward to seeing this paper in print and will certainly suggest it to my students in my graduate seminar on populism.

Author Response

Thank you very much for your positive feedback. In the revised version of the article I responded to two other reviewers who suggested adding some elements to the text and revising the conclusion with the main aim to elaborate more fully the generalizable finding of my article and what it contributes to the scholarly debate on populism.

Reviewer 3 Report

This is fine and timely contribution to scholarship. The article's exploration of the meaning of 'the people' is thorough and original, and it makes a worthwhile and compelling contribution to the existing theoretical literature on contemporary populism. I recommend publication without further revision. 

Author Response

(The authors gave the same response as above.)

Reviewer 4 Report

The present article is a well to read and informative piece about the change of the Lega Nord from a regionalist to an all-Italian party and the main enemy stereotypes evokes in its narratives in this change. Still, my main criticism would be: what is new? What is the particular contribution of this article that goes beyond already existing literature either? This should be clearly stated, be made more explicit. It is unclear which puzzle in particular the article wants to approach and explain, as also a clear research question is missing and the particular relevance of this article is not clearly articulated. From reading the paper the novelty of the arguments is not self-evident, as for the informed reader most is well known.

The key hypothesis of the article is that the way in which the foundational reference to the people is articulated and mobilized for strategic purposes has a significant effect on the populist political agenda. There are three aspects, which seem problematic for me: first, while the hypotheses is about the foundational reference to "the people" the thrust of the article is rather about the framing of opponents. Second, which findings would disprove your hypothesis, what would prove it? And third, the framing of "the people" or the "us vs. them" is the independent variable, while the political agenda (so the goals to be achieved) are the independent variable. Wouldn't it be more logical to be the other way round? Populist parties hold and develop specific goals and according to these goals they frame "a people" and the opponents of this people. As their agendas change, these framings change. And this is actually what the paper shows. Another critical aspect for me would also be that again the question of novelty would arise. This is a quite natural connection: what is new, what is the puzzle, why does this need closer analysis?

When reading the abstract, the interesting aspect was the question adressing the "disting ways in which nationalism and populism conceptualize and politically mobilize the notion of "people". But to really analyse this - and also to make generalizable hypotheses - a comparative research design would be necessary and would probably bring new insights.

Author Response

Thank you very much for your critical review and your suggestions. Let me briefly speak to the main points of your review and how I have addressed them in the revised manuscript.

First, what is the novelty of my article? You are surely right that the transformation of the Lega from a regionalist to a nationalist force is nothing new and has been noted by various scholars and political commentators in Italy. Yet, what I hope is innovative in my text is the attempt to analyze the transformation of the Lega as a case study for how populist parties/ movements use the reference to the people and provide it with meaning in their political campaigns. The reference to the ‘people’ in populist politics is a normal feature in the scholarly debate. Yet, an empirically grounded analysis of how the people are invoked in this actor’s discourse is rare. This empirical investigation allows me to make some generalizable observations about the difference of nationalist and populist mobilizing campaigns (the claim is summarized in lines 56-65).

On your point that my article is less on faming the ‘people’ than on its opponents. This was a deliberate decision recognizing the binary nature of collective identities and considering the deliberate vagueness of populist actors to frame the ‘people’. My hypothesis is that one can learn more about what populists depict as the ‘people’ by focusing on how their adversaries are described rather than by the discourse of what constitutes a ‘people’. In the revised version of the text, I have tried to make this conceptual approach more explicit.

 You are correct about the ambiguity regarding my ‘independent variable’. As I have elaborated (hopefully with more clarity) in the revised text I indeed see the framing of the people by populists and their political agenda as mutually dependent. In the manuscript, I added a segment in the conclusions to describe my interpretation of the empirical analysis (lines 490-497):

“From a conceptual perspective, the reliance on a mobilizing notion of the ‘people’ can be interpreted as an essential component of the populist political identity that is employed strategically to exploit political opportunities and, at the same time, as a key feature of the populist political narrative that shapes its political preferences. In comparison to nationalist actors, populists can rely on a broad repertoire of mobilizing the notion of the deprived ‘people’ and its enemies. Yet, once a populist actor has embarked on a particular story of the ‘people’s’ interests and identity, this narrative has its own structuring effects on the scope and mode of the populist political mobilization.”

I also added the following segment to the introduction:

“The hypothesis that I seek to develop based on the Lega as a case study is that, compared to nationalist modes of political mobilization, populism can effectively redefine its notion of the ‘people’ in response to emerging political opportunities. Yet, at the same time, strategically using a particular meaning and identity attributed to the ‘people’, populist actors redefine the scope within which they can mobilize their anti-elitist political creed.” (lines 92-97)  

This approach also allows me to interpret the Lega’s gradual adoption of a nativist discourse as integral to its reframed invocation of the ‘people’.   

On your last point: I agree a comparative research design could have allowed me to come to more substantiated generalizable findings. Yet, such a comparative approach was beyond the scope of this article. And I hope that, in its revised version, the manuscript has made a valid argument based on the case of the Lega.

Round 2

Reviewer 4 Report

the main criticisms have been addressed and it is now clearer what the article wants to achieve. I think it is a good study of the Lega Nord, I am still not fully convinced about the additional contribution to the literature, but this can probably be better achieved in future comparative studies.